# ZnT8 Loss of Function Mutation Increases Resistance of Human Embryonic Stem Cell-Derived Beta Cells to Apoptosis in Low Zinc Condition

**DOI:** 10.3390/cells12060903

**Published:** 2023-03-15

**Authors:** Lina Sui, Qian Du, Anthony Romer, Qi Su, Pauline L. Chabosseau, Yurong Xin, Jinrang Kim, Sandra Kleiner, Guy A. Rutter, Dieter Egli

**Affiliations:** 1Departments of Pediatrics, Naomi Berrie Diabetes Center, Obstetrics and Gynecology, Columbia Stem Cell Initiative, Columbia University Irving Medical Center, Columbia University, New York, NY 10032, USA; qd2139@cumc.columbia.edu (Q.D.);; 2Regeneron Pharmaceuticals, Inc., Tarrytown, NY 10591, USA; 3CR-CHUM, Faculté de Medicine, Université de Montréal, Montréal, QC H3T 1J4, Canada; 4Section of Cell Biology, Hammersmith Hospital, Imperial College, London WI2 ONN, UK; 5Lee Kong Chian School of Medicine, Nanyang Technological University, Singapore 308232, Singapore

**Keywords:** stem cell biology, stem cell, diabetes, type 2 diabetes, *SLC30A8*, zinc

## Abstract

The rare *SLC30A8* mutation encoding a truncating p.Arg138* variant (R138X) in zinc transporter 8 (ZnT8) is associated with a 65% reduced risk for type 2 diabetes. To determine whether ZnT8 is required for beta cell development and function, we derived human pluripotent stem cells carrying the R138X mutation and differentiated them into insulin-producing cells. We found that human pluripotent stem cells with homozygous or heterozygous R138X mutation and the null (KO) mutation have normal efficiency of differentiation towards insulin-producing cells, but these cells show diffuse granules that lack crystalline zinc-containing insulin granules. Insulin secretion is not compromised in vitro by KO or R138X mutations in human embryonic stem cell-derived beta cells (sc-beta cells). Likewise, the ability of sc-beta cells to secrete insulin and maintain glucose homeostasis after transplantation into mice was comparable across different genotypes. Interestingly, sc-beta cells with the *SLC30A8* KO mutation showed increased cytoplasmic zinc, and cells with either KO or R138X mutation were resistant to apoptosis when extracellular zinc was limiting. These findings are consistent with a protective role of zinc in cell death and with the protective role of zinc in T2D.

## 1. Introduction

Zinc transporter family (ZnTs) members are zinc exporters located on cell and organelle membranes and regulate the flow of zinc from the cytosol to granules and extracellularly. Zinc transporter 8 (ZnT8), encoded by *SLC30A8*, functions as a channel on insulin granules for transporting zinc ion into the granules to enable formation of stable insulin/zinc hexamers [1,2,3]. ZnT8 has been considered as an appealing target for preserving beta cell function in T2D because its loss of function mutations, including a pArg138* truncating variant, are associated with a more than 50% lower risk of developing T2D [4,5]. Several efforts have been made to study the mechanisms of the protective effect. In rodents, deletion of *Slc30A8 (SLC30A8)* globally or specifically in beta cells reduces intracellular zinc content, measured with Zinquin [6], dithizone [7], or with the recombinant cytosol-targeted probe, eCALWY4 [8,9], but does not affect cytosolic free Zn^2+^ measured by FluoZin-3, or insulin content [7,10,11,12,13]. These mouse models display variable alterations in glucose-stimulated insulin secretion (GSIS) and glucose metabolism; both increases and decreases in insulin secretion have been reported. A more recent mouse model carrying the human *SLC30A8* p.Arg138* allele shows enhanced insulin secretion under a high fat but not normal chow diet [14]. In recent studies by Dwivedi et al., heterozygous carriers of the protective p.Arg138* allele show greater insulin secretion during the oral glucose tolerance test (OGTT), but glucose excursions during a test meal do not differ between carriers of variant and wild type alleles [5]. Further studies using the human beta cell model EndoC-βH1, an immortalized beta cell line [15], by siRNA mediated knockdown of *SLC30A8* reveal reduced overall zinc content in beta cells as measured using the zinc-specific fluorescent dye Zinpyr-1, whereas no significant effect on GSIS was observed, though basal insulin secretion was increased [5]. Thus, studies in rodent models and immortalized human beta cell lines have revealed a range of phenotypes, and the consequences of ZnT8 deficiency for beta cell function and the mechanism of a protective effect of ZnT8 loss of function remain a matter of debate.

Zinc is an important trace element involved in numerous processes, including the regulation of apoptosis [2,16]. Reduced serum zinc levels have been detected in patients with type 2 diabetes (T2D), especially in those with poor glycemic control [17,18]. Additionally, plasma zinc is significantly decreased in older adults compared to that of young adults [19]. Furthermore, in mice on a limiting zinc diet, degeneration and apoptosis in the pancreas islet have been observed [20].

Here, we used human embryonic stem cell-derived beta cells (sc-beta cells) as a model system to understand the molecular and cellular consequences of ZnT8 loss of function by introducing a null mutation (KO), and homo-or heterozygous versions of p.Arg138* (R138X) into the *SLC30A8* locus. We demonstrate that ZnT8 deficient sc-beta cells display normal differentiation and function in vitro and in vivo, but reduced transport of zinc from the cytosol to the insulin granule, and experience lowered apoptosis under limiting zinc conditions.

## 2. Results

### 2.1. Human Embryonic Stem Cells with SLC30A8 Mutations Exhibit Normal Differentiation Potential towards Insulin-Producing Cells

To explore whether mutations in *SLC30A8* locus affect the development of insulin-producing cells, we first used CRISPR/Cas9 to introduce R138X and KO mutation into the *SLC30A8* locus of a human embryonic stem cell line MEL1, which carries an insulin-GFP reporter (Figure 1A). Two cell lines with KO mutation (KO23: c.65delA/p.Lys22fsX1 and KO2: c.63delC/p.Ala21AlafsX2) (Figure 1A and Appendix A) and two homozygous cell lines for R138X (C101: c.265C > T/c.265C > T and C121: c.265C > T/c.265C > T) (Figure 1A and Appendix A) were generated and verified by Sanger sequencing. All engineered cell lines displayed normal chromosome integrity determined by karyotyping (Appendix A). Differentiation efficiency towards insulin-producing cells was evaluated on wild type control (WT) (*n* = 5 independent replicates), R138X (*n* = 3 independent replicates for C101; *n* = 5 independent replicates for C121), and KO mutants (*n* = 3 independent replicates for KO2; *n* = 5 independent replicates for KO23). Upon differentiation, ~80% PDX1 and NKX6.1 double-positive pancreatic progenitors were generated by day 12 from all three genotypes (Figure 1B). On day 27, islet-like clusters were formed, and they expressed high levels of insulin as indicated by GFP expression (Figure 1C). Correspondingly, all cell lines stained positive for C-peptide and co-expressed NKX6.1, while KO and R138X sc-beta cells did not express detectable ZnT8 (Figure 1D). No significant difference in the percentage of C-peptide-positive cells was detected (WT: 61.8 ± 5.6%; KO2: 61.0 ± 5.3%; KO23: 57.6 ± 4.3%; C101: 60.0 ± 4%; C121: 59.4 ± 5.1%), and comparable percentages of C-peptide and NKX6.1 double-positive cells were observed out of the total C-peptide-positive cells from all five cell lines (WT: 54.8 ± 5.2%; KO2: 61.0 ± 5.0%; KO23: 60.4 ± 6.9%; C101: 63.3 ± 1.2%; C121: 60.2 ± 9.6%) (Figure 1D–F and Appendix A). Polyhormonal cells positive for C-peptide and glucagon were also detected in the derived islet clusters (WT: 25 ± 4.4%; KO23: 20 ± 5.3%; C121:19.7 ± 2.1%) (Figure 1D,G). MEL1 with an R138X heterozygous mutation (C89), as found in human subjects, differentiated into insulin-producing cells with comparable differentiation efficiency to WT, R138X, and KO mutants (Appendix A). Thus, loss of function mutations at the *SLC30A8* locus in either the homozygous or the heterozygous state do not impair beta cell differentiation from human pluripotent stem cells.

### 2.2. Sc-Beta Cells Carrying SLC30A8 Mutations Have Zinc-Depleted Secretory Granules

ZnT8 is responsible for transporting zinc into insulin granules, thus forming stable insulin-Zn^2+^ hexamers [22]. To investigate the consequences of ZnT8 deficiency, sc-beta cell clusters derived from all genotypes were stained with dithizone to assess the presence of zinc in the insulin granules [23]. WT sc-beta cell clusters stained positive with dithizone, whereas no strongly dithizone-positive cells were observed in sc-beta cell clusters derived from R138X and KO mutants. This result indicates that they were depleted of zinc in insulin secretory granules (Figure 2A). Next, the ultra-structure of insulin granules was examined in sc-beta cells after 8 months of transplantation using electron microscopy (Figure 2B). In the grafted cells after isolation, nearly 80% of the insulin granule in the WT sc-beta cells are crystalized with a dense core, whereas almost all insulin granules in R138X (99.69 ± 0.62%) and KO (97.97 ± 0.86%) displayed an enlarged and diffused granule core (Figure 2D). The degree of granule density, indicated by the grayscale value (black = 0, white = 250), was also analyzed. Granules that are denser and darker are presented with lower grayscale value. It was found that insulin granules in KO (113.25 ± 27.23%) and R138X (78.29 ± 33.80%) were both significantly less dense with higher grayscale value than that in WT (35.86 ± 30.85%) cells (Figure 2E). Interestingly, granules in the KO sc-beta cells are significantly more diffused than those of the R138X mutant sc-beta cells. The diffused granules were also observed in mouse islets segregating for the KO and R138X transgenic mice (mice generation described previously in [14]). In mice, insulin granules displayed no dark cores in the explanted mouse pancreas with R138X mutation under the normal chow diet and with KO mutation under the high fat diet. Whereas, in WT control mice, a dark and dense core was formed in each insulin granule under the normal chow diet. (Figure 2C).

### 2.3. Transcriptome Analysis Shows Discrepant Expression of Genes Involved in Zinc Ion Homeostasis in SLC30A8 Mutants

We next examined the effects of *SLC30A8* mutants on gene expression in sc-beta cell clusters on day 27 of differentiation. Based on a profile of cell specific markers, 10 endocrine cell populations and 1 non-endocrine population using cells from all conditions were identified (Figure 3A and Appendix A). All identified endocrine populations expressed a high level of CHGA and different level of insulin with the highest expression in identified sc-beta cells (Appendix A). Note that among these populations, SC-beta.PP cells are characterized by expressing both SC-beta cell genes and SC-PP cell genes, and SC-beta.zinc cells express not only SC-beta cell genes but also express high levels of SLC39A5 and metallothionine genes. Other endocrine cells have been distinguished based on the lineage specific markers, but their co-expression with insulin indicated that they were polyhormonal (Appendix A).

Pancreatic beta cells and alpha cells are major cell types that express *SLC30A8* [10,24]. First, the percentage of sc-beta and sc-alpha cells was quantified in WT, KO, R138X derived clusters (Figure 3B). We found that the introduced mutations in *SLC30A8* did not alter the percentage of sc-beta cells and sc-alpha cells in the sc-islet like clusters (Figure 3C), indicating that *SLC30A8* is not essential for the fate determination of pancreatic endocrine cells. Double hormone positive cells, which have been shown to become sc-alpha cells [25], are also not different between genotypes.

Next, genes differentially expressed between mutants and WT were identified in sc-beta cells (Figure 3D). KO and R138X shared similar gene profiles, and both were distinct from the gene profile of the WT. As expected, the expression of *SLC30A8* together with metallothionein genes involved in the maintenance of zinc ion homeostasis were consistently downregulated in R138X and KO mutants (Figure 3E). Metal responsive transcriptional factor 1 (MTF-1) is a zinc dependent transcription factor regulating the transcription of genes involved in zinc transport and chelation. It also regulates the transcription of metallothionein which depends on zinc ions [26]. The lower expression of MT genes in both KO and R138X mutant sc-beta cells indicates the intracellular disruption of zinc homeostasis (Figure 3E). In contrast, genes involved in insulin-containing vesicles and insulin secretion were upregulated in R138X and KO beta cells (Figure 3F). Taken together, these results indicated that the loss of function KO and R138X mutations cause relatively minor transcriptional changes overall compared to wild type cells. Those changes that are shared in both KO and R138X point to possible molecular mechanisms through which pancreatic endocrine function may be altered: control of insulin hormone secretion and zinc ion homeostasis, each of which we examine separately in the following chapters.

### 2.4. Insulin Secretion and Glucose Regulation of SLC30A8 Mutant Sc-Beta Cell Are Comparable to WT

As we noted an enrichment in the expression of hormone secretion-related genes in both KO and R138X sc-beta cells (Appendix A), we next sought to address whether loss of function mutations in the *SLC30A8* locus have an effect on insulin secretion and beta cell function. Sc-beta cell clusters derived from R138X mutant (*n* = 15 including 11 mice transplanted with clone C121 from 3 batches of independent differentiation and 4 mice with clone C101 from 1 batch of differentiation), KO mutant (*n* = 13 including 9 mice with KO23 from 3 batches of independent differentiation and 4 mice with KO2 from 1 batch of differentiation), and WT (*n* = 14 from 4 batches of independent differentiation) cell lines with equivalent number of ~2 million cells were transplanted into the leg muscle of immunodeficient mice.

Human C-peptide secretion in the fed state was monitored to evaluate insulin production of grafted sc-beta cells after transplantation. Human C-peptide secretion increased over time in mice transplanted with cell clusters of WT and mutants (Figure 4A). We compared human C-peptide and insulin secretion at a 6-month time point after transplantation, and no significant differences among WT, KO, and R138X cell lines were noticed (Figure 4B,C). Interestingly, mouse C-peptide was suppressed when human C-peptide was secreted at a high level, even prior to the application of streptozotocin (STZ) (Figure 4D). This shows that the human stem cell derived grafts functionally take over blood glucose regulation from the endogenous mouse pancreas.

To examine the function of sc-beta cells after 26 weeks of maturation in vivo, mice were treated with streptozotocin to eliminate endogenous mouse beta cells. Mouse beta cells were successfully eliminated, as confirmed by barely detectable levels of mouse C-peptide in blood (Appendix A). Overall, 14 mice transplanted with 2 R138X cell lines, 11 mice transplanted with 2 KO cell lines, and 10 mice transplanted with a WT cell line could maintain glucose homeostasis after mouse beta cells were ablated (Appendix A). We also transplanted 2 mice with an R138X heterozygous mutation (C89) cell line, and they successfully maintained normal glucose levels after STZ treatment while the non-graft mice experienced hyperglycemia after the same treatment (Figure 4E and Appendix A).

Glucose tolerance tests (ipGTT) were performed on the mice with normal levels of blood glucose after STZ treatment. Mice transplanted with sc-derived clusters of all four genotypes, including WT, KO, R138X homozygous and heterozygous *SLC30A8* mutant cells, were able to normalize blood glucose (Appendix A) after glucose injection with increased insulin secretion at comparable levels (Appendix A). We noticed that both R138X and KO mutants tended to clear blood glucose more efficiently after the glucose challenge, although the differences compared to WT cell-bearing animals were not statistically different (Appendix A). We then repeated the ipGTT assay by including a 15 min measurement of blood glucose. With the increased mice number in each group, we found that mice transplanted with KO cell lines exhibited reduced glucose levels at both 15 min and 30 min (Figure 4F). Mice with R138X transplants also showed lower 15 min glucose level than that of the WT, though this difference was not significant. In terms of overall glucose regulation, we analyzed the area under the curve (AUC) integrated from 0 min to 60 min and observed significantly lower glucose excursion and more tolerance in the KO compared to the WT (Figure 4G).

Noting that, consistent with a modest increase of insulin secretion, we found an increase of insulin secretion in vitro when cells were incubated with 2 mM glucose solution for 1 h compared to WT control (Appendix A). This difference was not due to variability in insulin content, which was comparable among KO, R138X, and WT (Appendix A).

To rule out the possibility that differences in insulin secretion in vivo may be caused by alterations in the number of insulin-producing cells for each cell line, we measured the size of the whole graft in vivo (Appendix A) and quantified the insulin-GFP positive area in the graft after isolation using bioluminescence imaging (Appendix A). We observed similar graft size within the region of interest (ROI) in the GFP-positive area from each mouse grafted with WT, KO, and R138X sc-beta cells (Appendix A). Insulin expression was confirmed in the grafted cells after isolation, and the grafted cells formed islet-like structures containing monohormonal insulin and glucagon-positive cells (representative pictures for KO and R138X are shown in Figure 4H). This shows that the grafting of cells is not negatively affected by the loss of ZnT8 function.

To determine if the modest differences in ipGTT or insulin secretion are functionally meaningful in regulating blood glucose levels, HbA1C levels were measured in mice transplanted with sc-beta cells derived from WT and mutants. Regardless of the genotypes of sc-beta cells, all transplanted mice could maintain normal HbA1C levels. As a result, no apparent improvement in blood glucose regulation in the KO was noticed (Figure 4I), and all these transplanted mice have significantly lower HbA1C level than mice without any graft.

To determine if proinsulin processing is altered, we examined the proinsulin and insulin ratio in R138X, KO, and WT sc-beta cells before and after transplantation. Each line demonstrated a comparable proinsulin to insulin production ratio (Appendix A and Figure 4J), indicating that proinsulin processing was comparable to WT in *SLC30A8* mutants. We also calculated insulin/C-peptide and proinsulin/insulin content ratio in the isolated grafted cells and observed no difference in the ratio between WT, KO, and R138X sc-beta cells (Appendix A).

In summary, we conclude that insulin processing, secretion and sc-beta cell function are not compromised due to the mutation introduced in the *SLC30A8* locus.

### 2.5. mTOR Activity Is not Influenced by SLC30A8 Expression

We found that a panel of ribosome genes were significantly down in *SLC30A8* mutants in the sc-beta population compared to WT in single cell RNA-seq analysis (Appendix A). Ribosome biogenesis is associated with the mTOR signaling pathway [28]. Intracellular free zinc has been shown to correlate with the mTOR signaling pathway through phosphorylation of p70 S6 kinase [29]. To determine if reduced expression of ribosome genes in R138X and KO is due to the decreased mTOR activity, we isolated insulin-positive cells based on GFP expression with flow cytometry and evaluated the phosphorylation of mTOR, downstream effectors S6 kinase and S6 ribosome protein by Western blot. WT and mutant cells expressed similar levels of phosphorylated form of mTOR, S6 kinase, and S6 ribosome from the total form of individuals (Appendix A). These data suggest that mTOR activity is not affected by the expression of *SLC30A8*, contrasting with the knockdown of ZnT8 in EndoC-betaH1 cells which increased mTOR activity [5].

### 2.6. SLC30A8 Mutant Sc-Beta Cells Are Resistant to Apoptosis in Low Zinc Conditions

Transcriptome analysis as well as the reduction of crystalline insulin granules establish a disruption of intracellular zinc homeostasis in mutant cells, consistent with the molecular function of *SLC30A8*. To further explore whether free cytosolic Zn^2+^ concentrations are altered in *SLC30A8* mutants, FRET (fluorescence resonance energy transfer)-based cytoplasmic free Zn^2+^ content analysis [30] was performed on insulin GFP-positive cells derived from R138X homozygous, R138X heterozygous, KO, and WT cell lines. The maximum and minimal ratios were, respectively, obtained upon intracellular zinc chelation with Tetrakis-(2-pyridylmethyl) ethylenediamine (TPEN) and zinc saturation with ZnCl_2_ in the presence of the Zn^2+^ ionophore, pyrithione (Figure 5A). Cytosolic zinc levels in *SLC30A8* mutant sc-beta cells were comparable to or higher than those in WT cells. Notably, free cytosolic zinc was significantly elevated in cells carrying the KO mutation compared to other cell lines (Figure 5B). The difference between KO mutant and R138X mutant may be due to the expression of a partially-active *SLC30A8* protein which retains two transmembrane domains in R138X mutant [14] (Figure 1A). A functional difference between the two alleles was also suggested by the higher percentage of diffuse insulin granules detected in the KO than in the R138X (Figure 2E).

Zinc is a regulator of programmed cell death required for cell survival, and its depletion induces cell death (reviewed in [31,32]). Normally, there are around 0.47% of beta cells that undergo apoptosis in control non-diabetic human islets [33]; while, in autopsies of T2D islets, there are an average of ~7% beta cells that undergo apoptosis [34]. To investigate whether the elevated cytosolic zinc in *SLC30A8* mutants can protect cells from apoptosis after short-term extracellular zinc depletion, extracellular zinc provided by the culture medium was reduced by 5 µM TPEN, a chelator that has a high affinity for zinc and removes free zinc in the medium, and cell apoptosis was examined by staining the untreated and TPEN-treated cells with TUNEL. After 48 h of treatment, the percentage of TUNEL-positive cells was significantly increased after TPEN treatment compared to untreated control in WT (WT: 3.71 ± 2.65% with TPEN; 0.52 ± 1.17% without TPEN). In contrast, in *SLC30A8* mutants, no significant increase in apoptotic cells was detected after TPEN treatment (KO: 0.64 ± 0.82% with TPEN; 0.14 ± 0.55% without TPEN) (R138X: 0.80 ± 0.95% with TPEN; 0.28 ± 0.63% without TPEN) (Figure 5C,D).

Taken together, the above findings reveal that both R138X and KO loss of function mutations in *SLC30A8* impair the crystallization of insulin due to the failure of channeling zinc into insulin granules. This can result in an elevation of free cytoplasmic zinc in the latter case, conferring resistance to apoptosis after extracellular zinc depletion.

## 3. Discussion

In this study, we used human embryonic stem cells as a model to investigate the differentiation and function of *SLC30A8* p.Arg138* variant, a patient mutation that is protective against T2D. Our results demonstrate that loss of function mutations in *SLC30A8* locus do not influence differentiation toward the pancreatic lineage and to insulin-producing cells. This finding is consistent with observations in *SLC30A8* KO mice which have normal beta cell mass and function [7,10,14]. A previous study has reported low differentiation efficiency of induced pluripotent stem cells (iPSCs) carrying mutations in *SLC30A8* to insulin-producing cells [5]. However, this phenotype is not readily reconciled with the protective effect of the mutation, and the molecular mechanisms that might be responsible are not known. The role of other confounding factors in this discrepancy cannot be excluded, as iPSC lines can show variable differentiation competence [35]. The effect of *SLC30A8* mutations on beta cells was evident by a lack of dense-core zinc positive insulin granules. Zinc is essential for the formation of crystalized insulin with dense core granules and regulated insulin secretion. Our study shows that both complete loss of function mutations (KO) and mutations in p.Arg138* variant (R138X) both possess reduced crystalized insulin granules and reduced granule zinc content while sc-beta cells from KO has lost crystalized granule more completely.

A recent study showed that genes related to beta cell maturation are upregulated in *SLC30A8* KO sc-derived beta cells [36], providing a potential mechanism for the protective effect of *SLC30A8* mutations. In our scRNA-seq, we also observed upregulation of *PCSK2, ONECUT2,* and *IAPP* in *SLC30A8* KO cells, concordant with reported findings [36]. Nevertheless, in our human embryonic stem cell model of ZnT8 loss of function mutations, R138X and KO sc-beta cell clusters were neither compromised nor improved in insulin processing or glucose-stimulated secretion. A previous study has shown that after transplantation, sc-beta cells are essentially equivalent to mature pancreatic beta cells [37], which is why we chose cell grafting into mice for functional testing of insulin secretion. After transplantation in vivo, the response to glucose of cells derived from R138X and KO stem cells is comparable with WT. In the ipGTT functional test, we observed a small improvement of glucose clearance in mice with *SLC30A8* KO mutant at both 15 and 30 min after glucose injection. However, analysis of HbA1C levels did not reveal differences between mice dependent on glucose regulation by grafts of WT or *SLC30A8* mutant cells; thus the functionally protective relevance of a potentially improved insulin secretion was not evident over the time frame of the mouse experiments performed here. Improved insulin secretion profiles were reported in a prior study [cite reference]. Our studies are not inconsistent with this report, but any effect is very small. The increased expression of genes involved in beta cell maturation or insulin secretion observed here as well as in a prior study (REF), is not inconsistent with functional improvements, but may also be a compensatory response to the uncrystallized insulin granules.

We thus also evaluated other mechanisms of a protective effect of *SLC30A8* mutations on beta cell function based on changes to zinc homeostasis. We find that cytoplasmic free zinc content remains the same in R138X, and is higher in KO cells than in controls. This indicates that loss function of ZnT8 influences the transportation of zinc from the cytoplasm into insulin granules, and that the disruption of transport of cytosolic zinc to the granule can increase the availability of cytosolic zinc. We find that the consequences of ZnT8 loss of function lead to significantly more resistance to cell apoptosis when zinc in culture media is reduced through chelation. A recent study also found that isolated islets from *SLC30A8* KO mice are protected against hypoxia-induced cell death when compared to wildtype islets [38]. Zinc supplement was reported to have protective modulation against high glucose induced apoptosis in renal tubular epithelial cells [16]. Wild type mice on a limiting zinc diet show beta cell apoptosis and degeneration of pancreatic endocrine function [20]. Importantly, zinc has been implicated in T2D: hypozincemia was observed in patients with diabetes [17,18,39,40]. As serum zinc level decreases in the aging process [19,41], zinc deficiency may contribute to the rise in T2D incidence during aging. Our findings suggest that the lack of granular zinc may be protective, as it increases zinc availability in other cellular compartments. Whether ZnT8 deficiency protects against apoptosis under physiological circumstances in grafted mice as well as in patients remains to be determined.

Ma et al. provided an alternative interpretation for the protective effect of ZnT8 in a stem cell model: an inhibitory effect of extracellular zinc on insulin secretion [36]. This zinc would only be released from zinc-containing insulin granules, not from ZnT8 mutant beta cells. However, experimental treatments involved very high concentrations of zinc (100 µM), and 100 µM extracellular zinc has a long-term toxic effect on glucose-stimulated electrical activity of pancreatic beta cells [42]. Lower concentrations of 10 µM have no significant effect [43]. The authors also showed that insulin knockout cells, but not *SLC30A8* mutant cells, had an inhibitory effect on insulin secretion when aggregated with wild type beta cells. However, this would require the formation of granules without insulin, but with high zinc levels. Insulin granules form after proinsulin is produced and packed into granular vesicles [44]. Granules form through inclusion of the secreted protein, rather than through inclusion of metal ions alone. Hollow secretory granules without insulin but loaded with high zinc was not demonstrated, and how secreted zinc was measured has not been clarified in the study. Thus, whether the absence of zinc from secreted granules mediates the protective effect of ZnT8 loss of function is not clear.

A limitation of this study is that no functional consequences of the heterozygous *SLC30A8* mutation are apparent, neither in vitro nor in vivo. The consequences in homozygous mutant cells are surprisingly small, but consistent with a protective effect against T2D. Conclusions based on homozygous mutations assume that the effects of heterozygous mutations have the same directionality as a homozygous mutation, and that homozygous mutant cells can serve as an exaggerated model. It may be that the functional consequences of heterozygous loss of function mutations become apparent only after years of metabolic stress. Of note, the effect of the common T2D-associated *SLC30A8* risk variant rs13266634 (R325W) on ZnT8 activity is likely to be a gain-of-function [45].

In addition, currently unknown are the molecular mechanisms for species-specific differences and the variability in phenotypic expression of ZnT8 deletion in mice. Interestingly, the inactivation of ZnT8 in mouse beta cells was reported to result in lowered free cytosolic Zn^2+^ levels, as measured using the same strategy as used here (the eCALWY GFP-based probe) [8,9], whereas human *SLC30A8* KO sc-beta cells displayed the same or elevated cytosolic Zn^2+^. Loss of ZnT8 function in mice causes impaired glucose tolerance or has no effect [3], whereas in humans, rare loss of function variants offer protection against T2D [4,5]. Differences and potential variability in the effect on cytoplasmic zinc may thus underly these species-specific differences.

In summary, our studies show upregulated cytoplasmic free zinc in the *SLC30A8* KO sc-beta cells, and a protective effect on beta cell survival. We also demonstrate that ZnT8 is dispensable for both the generation and function of human stem cell-derived beta cells. Inferring from our findings, the protective effect of ZnT8 loss of function for T2D could potentially be substituted through a readily available zinc supplement. Consistent with this view, common *SLC30A8* variants influence the protective effect of dietary zinc supplementation on T2D risk in humans [46,47,48]. Adding further relevance to this study, loss of ZnT8 function may facilitate the generation of improved and more apoptosis resistant stem cell derived beta cell grafts for different forms of diabetes, including for T1D where ZnT8 is an important autoantigen [21,49].

### Note Added in Proof

While this study was under consideration for publication, another study reported normal differentiation of ZnT8 loss of function stem cells to beta-like cells [36].

## 4. Methods

### 4.1. Human Pluripotent Stem Cell Culture and Gene Editing with CRISPR/Cas9

Human pluripotent stem cells were cultured and maintained on feeder-free plates with StemFlex Medium (Cat. No. A3349401, Thermo Fisher Scientific, Waltham, MA, USA) as described [50]. We established cell lines with mutations by introducing homozygous p.Arg138* mutation and KO mutations into human embryonic stem cell line MEL1 (NIH registry #0139) with CRISPR/Cas9. This cell line carries in insulin-GFP knockin [51]. Guide RNA was designed by following the published protocol and synthesized by Integrated DNA Technologies (IDT) (https://www.idtdna.com/pages (accessed on 1 January 2023). A 120 bp repair ssDNA template with the desired sequence was synthesized by IDT (guide sequences and repair template sequence listed in Appendix A). Cas9-GFP plasmid was obtained from Addgene (Cat. No. 44719). Then, 2.5 µg of each component were transfected into 1 million cells with LONZA nucleofector. Normal karyotypes for all cell lines were validated by Cell Line Genetics (Appendix A).

### 4.2. Insulin-Producing Cell Differentiation from Human Pluripotent Stem Cells

Insulin-producing cells were differentiated from human pluripotent stem cell lines at certain passages, ranging from 25–30 passages, using the published protocol [50], with aphidicolin treatment to obtain better differentiation and transplantation results [35]. First, cells were cultured for 4 days using STEMdiff™ Definitive Endoderm Differentiation Kit (Cat. No. 05110, STEMCELL Technologies, Vancouver, BC, Canada) for definitive endoderm induction. Primitive gut tube was induced by RPMI 1640 plus GlutaMAX (Cat. No. 61870-127, Life Technology, Carlsbad, CA, USA) + 1% (*v/v*) Penicillin–Streptomycin (PS) (Cat. No. 15070-063, Thermo Fisher Scientific, Waltham, MA, USA) + 1% (*v/v*) B-27 Serum-Free Supplement (50× (Cat. No. 17504044, Life Technology, Carlsbad, CA, USA) + 50 ng/mL FGF7 (Cat. No. 251-KG, R&D System, NE Minneapolis, MN, USA) from day 4 to day 6. Posterior foregut is induced by DMEM plus GlutaMax (DMEM) (Cat. No. 10569-044, Life Technology, Carlsbad, CA, USA) with 1% (*v/v*) PS + 1% (*v/v*) B-27 + 0.25 μM KAAD-Cyclopamine (Cat. No. 04-0028, Stemgent, Cambridge, MA, USA) + 2 μM Retinoic acid (Cat. No. 04-0021, Stemgent, Cambridge, MA, USA) + 0.25 μM LDN193189 (Cat. No. 04-0074, Stemgent, Cambridge, MA, USA) from day 6 to day 8. We then changed medium to DMEM + 1% (*v/v*) PS + 1% (*v/v*) B-27 + 50 ng/mL EGF (Cat. No. 236-EG, R&D System, NE Minneapolis, MN, USA) + 25 ng/mL FGF7 for 4 days to generate pancreatic progenitor cells. On day 12, cells were dissociated into single cells using TrypLE^TM^ Express (Cat. No. 12605036, Life Technologies, Carlsbad, CA, USA) and clustered in AggreWell 400 6-well plates (Cat. No. 34425, STEMCELL Technologies, Vancouver, Canada) or ultra-low attachment 96-well plates (Cat. No. 07-201-680, Thermo Fisher Scientific, Waltham, MA, USA) with DMEM + 1% (*v/v*) PS + 1% (*v/v*) B-27 + 1 μM ALK5 inhibitor (Stemgent, cat. No. 04-0015) + 10 μg/mL heparin (Cat. No. H3149, Sigma-Aldrich, Burlington, MA, USA) + 25 ng/mL FGF7 + 10 μM Y-27632, ROCK inhibitor. On day 13, newly formed clusters were transferred into ultra-low attachment 6-well plates (Cat. No. 07-200-601, Thermo Fisher Scientific, Waltham, MA, USA) (or remaining culture in the 96-well ultra-low attachment plates), and medium was changed to RPMI 1640 plus GlutaMAX + 1% (*v/v*) PS + 1% (*v/v*) B-27 + 1 μM thyroid hormone (T3) (Cat. No. T6397, Sigma-Aldrich, Burlington, MA, USA) + 10 μM ALK5 inhibitor + 10 μM zinc sulfate (Cat. No. Z4750, Sigma-Aldrich, Burlington, MA, USA) + 10 μg/mL heparin + 100 nM gamma-secretase inhibitor (DBZ) (EMD Millipore, cat. No. 565789) + 10 μM Y-27632, ROCK inhibitor for 7 days. From day 20 to day 27, to induce pancreatic beta cell, medium was changed to RPMI 1640 plus GlutaMAX + 1% (*v/v*) PS + 1% (*v/v*) B-27 + 10% (*v/v*) fetal bovine serum (Cat. No. S11150, Atlanta Biologicals, Flowery Branch, GA, USA). From day 15 to day 27, besides the component mentioned above, 2 μM aphidicolin (Cat. No. A0781-1MG, Sigma-Aldrich, Burlington, MA, USA) was added to the medium for differentiation improvement [35]. Note that the MEL1 cell line only has one insulin allele, but it is sufficient in blood glucose regulation as shown in all of our functional studies in vivo.

### 4.3. Dithizone Staining

Dithizone stock solution (1 mg/mL) was prepared by dissolving dithizone (Cat. No. D5130, Sigma-Aldrich) into DMSO and storing at −20 °C. Stem cell derived beta-like cell clusters were stained with dithizone by adding 50 µL of dithizone stock solution into 1 mL of medium to reach a final concentration of 50 µg/mL, and this was incubated for 1 min. Pictures were taken with an OLYMPUS IX73 microscope.

### 4.4. Immunocytochemistry

At 27 days of differentiation, clusters were fixed with 4% paraformaldehyde (PFA) at room temperature (RT) for 10 min. Grafts taken from the mice were also fixed with 4% PFA at RT for 1 h. The following steps were performed according to the published method [50]. Primary antibodies are listed in Appendix A, and secondary antibodies are listed in Appendix A. Pictures were taken with an OLYMPUS IX73 fluorescent microscope or ZEISS LSM 710 confocal microscope. For each panel, pictures were taken with same exposure time.

### 4.5. Flow Cytometry

The beta-like cell clusters were treated with TrypLE^TM^ Express (Cat. No. 12605036, Life Technologies) into single cells. Then, they were fixed with 4% PFA for 10 min and permeabilizated with cold methanol at −20 °C for 10 min. Primary antibodies were added at a dilution of indicated ratio in Appendix A in phosphate buffered saline (PBS) containing 0.5% BSA at 4 °C for 1 h. Secondary antibodies were added accordingly (Appendix A) at a dilution of 1:500 at room temperature for 1 h. The cells were then filtered with BD Falcon 12 mm × 75 mm tube with a cell strainer cap prior to flow cytometry analysis.

### 4.6. Single-Cell RNA Sequencing and Read Mapping

A total of 21,273 cells (WT: 8648; KO: 6397; R138X: 6228) were sequenced. Preparation of the cells was performed according to methods previously described [27]. Briefly, single cells were suspended in PBS + 0.04% BSA and for cell hashing, totalseq-A anti-human hashtag antibodies (BioLegend) were used. Samples from different groups were individually stained with one of the hashtag antibodies and washed three times. Twelve samples for each were pooled at equal concentrations, and the pool was loaded into the 10X Chromium instrument at 32,000 cells per lane. Single-cell RNA-seq libraries were prepared using Chromium Single Cell 3′ Reagent Kits v2 (10X Genomics). Hashtag libraries were generated as described in [52]. Illumina NextSeq500 was used for sequencing. Cell Ranger Single-Cell Software Suite (10X Genomics, v2) was utilized for sequence alignment and quantification of expression. Reads were aligned to the B37.3 Human Genome assembly and UCSC gene model.

### 4.7. Single-Cell Data Analysis

Single cell data analysis was performed as previously described [35]. Cell-hashing tags were demultiplexed using HTODemux function (Seurat v3). We excluded empty droplets and doublets. The exclusion criteria were as follows: (1) total UMI bigger or smaller than three folds of median absolute deviation (MAD); (2) detected genes bigger or smaller than 3 folds of MAD; (3) detected genes in the first of the bimodal distribution (classified by mclust); (4) mitochondrial gene ratio bigger than 0.15. Cells from three experiment groups (WT, KO, and R138X) were integrated and clustered to identify cell types and subpopulations (Seurat v3).

### 4.8. Static Glucose Stimulated Insulin Secretion

Krebs Ringer buffer (KRB) was prepared by addition of 129 mM NaCl, 4.8 mM KCl, 2.5 mM CaCl_2_, 1.2 mM MgSO_4_, 1 mM Na_2_HPO_4_, 1.2 mM KH_2_PO_4_, 5 mM NaHCO_3_, 10 mM HEPES and 0.1% BSA in deionized water and was sterilized with a 0.22 μm filter. The 2 mM glucose solutions were prepared in KRB for low glucose challenge of sc-beta cell clusters. Then, 10–20 sc-beta cell clusters (~5 × 10^5^ cells) were collected from WT, KO, and R138X and pre-incubated in 500 μL 2 mM glucose solution for 1 h. Clusters were then washed once with 2 mM glucose solution and subsequently incubated in 200 µL of 2 mM glucose for 1 h. Finally, 130 μL supernatant from each condition was collected. Cell clusters were centrifuged down and resuspended in 50 µL high salt buffer and sonicated for protein content preparation and DNA measurement with Nano Drop Spectrophotometer ND-1000. Protein content and supernatants in 2 mM glucose solution were processed using Mercodia Insulin ELISA kit (Cat. No. 10-1113-01, Mercodia, Uppsala, Sweden).

### 4.9. Transplantation and In Vivo Assay

The 8–10 weeks old male immunocompromised mice (NOD.Cg-Prkdcscid Il2rgtm1Wjl/SzJ (NSG) from Jackson laboratories, Cat. No. 005557) were used for transplantation. For intra-leg muscle transplantation, ~2 million cells were collected and transferred to a tube with 50 μL Matrigel (Cat. No. 354277, Fisher Scientific). Cluster injections in the leg muscle were performed using a 21G × 11/2” needle (Cat. No. 305177). The human C-peptide levels in mouse serum were measured every two weeks in the fed state. For analysis of glucose stimulated insulin secretion, at 2 weeks after mouse beta-like cells were ablated with one high dose (150 mg/kg) of streptozotocin (Cat. No. S0130-1G, Sigma-Aldrich), intraperitoneal glucose tolerance test (ipGTT) was performed by fasting overnight (for 16 h) with bedding change and injecting 2 g/kg D-(+)-glucose (Cat. No. G8270, Sigma-Aldrich, Burlington, MA, USA) solution. Blood was obtained by clipping tail tips of the mice and was collected using 75 mm heparinized capillary tubes (Cat. No. 1-000-7500-HC/5, Drummond Scientific, Broomall, PA, USA), then transferred to heparin-coated tubes (Cat. No. 022379208, Eppendorf, Hamburg, Germany) at fed state, at fasting state, and 30 min after glucose injection. Plasma was collected by centrifuging heparin-coated tubes at 3000× *g* for 15 min at 4 °C. The supernatants were collected for C-peptide and insulin detection with Mercodia Insulin ELISA kit (Cat. No. 10-1113-01, Mercodia, Uppsala, Sweden), Mercodia Ultrasensitive C-peptide ELISA (Cat. No. 10-1141-01, Mercodia, Uppsala, Sweden), and Mercodia Proinsulin ELISA (Cat. No. 10-1118-01, Mercodia, Uppsala, Sweden). Blood glucose levels were measured by a glucometer (FreeStyle Lite, Abbott, Chicago, IL, USA), using a drop of blood obtained from clipping mice tail tips at fed state, fasting state, 15min, and every 30 min after glucose injection for 2 h. All animal protocols were approved by the Institutional Animal Care and Use Committee of Columbia University.

### 4.10. Quantification of Zinc Concentration Using FRET Sensor eCALWY-4

The dual read-out FRET-based assays for intracellular zinc measurement were performed as described in the previous literature [53]. Cell clusters were infected with an adenovirus construct for the genetically encoded zinc sensor eCALWY-4 and let to express for 24 h. Before imaging, cells clusters were dissociated using accutase for 5 min at 37 °C and dispersed by pipetting up and down. Cells were then pelleted, resuspended in normal media, and let to attach for 3 h on glass slides treated with polylysine; then, zinc imaging was performed as previously described [54]. Cells on coverslips were washed twice in Krebs-HEPES-bicarbonate (KHB) buffer (140 mM NaCl, 3.6 mM KCl, 0.5 mM NaH_2_PO_4_, 0.2 mM MgSO_4_, 1.5 mM CaCl_2_, 10 mM Hepes, 25 mM NaHCO_3_), which was warmed, bubbled with 95% O_2_: 5% CO_2_, set to pH 7.4, and contained 11mM glucose. Cells were then transferred in an imaging chamber and maintained at 37 °C throughout with a heating stage (MC60, LINKAM, Scientific Instruments, Redhill, UK), and KHB buffer was perfused (1.5 to 2 mL/min). Images were captured at 433 nm monochromatic excitation wavelength (Polychrome IV, Till photonics) using an Olympus IX-70 wide-field microscope with a 40×/1.35NA oil immersion objective and a zyla sCMOS camera (Andor Technology, Belfast, UK) controlled by Micromanager software. Acquisition rate was set at 20 images/minute. Emitted light was split and filtered by a Dual-View beam splitter (Photometrics) equipped with a 505dcxn dichroic mirror and two emission filters (Chroma Technology—D470/24 for cerulean and D535/30 for citrine).

### 4.11. TPEN Treatment and Quantification of TUNEL/C-Peptide Double Positive sc-Beta Cells

On day 27 of differentiation, clusters were treated with 5 μM of TPEN for 48 h. Preparations for cryosection were performed following the published method [50] and the above. For TUNEL/immuno-double staining, we followed the protocol described previously [55]. For the quantification of TUNEL/c-peptide double positive cells out of total c-peptide positive cells per cluster section, we performed manual counting using ZEISS Blue Edition.

### 4.12. Image Analysis of Fluorescence Emission Ratio

Image analysis process was described previously [54]. Briefly, ImageJ software was used with a designed macro and after subducting the background, the fluorescence emission ratios were derived. Steady-state fluorescence intensity ratio citrine/cerulean was evaluated. Next, maximum and minimum ratios were measured for the calculation of free cytosolic Zn^2+^ concentration. The value of the concentration was determined using the following formula: [Zn^2+^] = K_d_ × (R_max_ − R)/(R − R_min_). In the formula, the maximum ratio (R_max_) was achieved when intracellular zinc chelated with 50 μM TPEN. Additionally, the minimum ratio (R_min_) was achieved when adding 100 μM ZnCl_2_ with Zn^2+^ ionophore, pyrithione (5 μM) resulted in reaching zinc saturation.

### 4.13. In Vivo/In Vitro Graft Imaging and Graft Content Measurement

*SLC30A8* mutant and wildtype MEL1 lines possess GAPDH^Luciferase/wt^ and INS^GFP/wt^ double reporter. Before bioluminescence and fluorescence imaging, the NSG mice transplanted with SLC30A8 mutant and wildtype grafts were i.p injected with 150 mg/kg body weight of D-luciferin potassium salt (Gold Biotechnology, luck-2G) in PBS at least 15 min before imaging on a IVIS spectrum optical imaging system (PerkinElmer) (previously described in [35]). Signals were acquired with 1 min exposure and analyzed using the Living image analysis software (Xenogen Corp. version 4.0). Equal sizes of regions of interest (ROI) were drawn for all graft imaging in the same panel. Signal in the left thigh and photons emitted over the time of exposure within the ROI were measured. Luminescence was measured as described for bioluminescence after isolation of the grafts. Background signals were subtracted from a nearby region. For the graft insulin/proinsulin content measurement, we first isolated GFP positive parts from the leg based on in vitro imaging. Individual graft pieces were snap frozen in liquid nitrogen and homogenized into powder using a mortar and pestle. Then, 1mL of complete extraction buffer (100 mM Tris, pH 7.4 + 150 mM NaCl + 1 mM EGTA + 1 mM EDTA + 1% Triton X-100 + 0.5% Sodium deoxycholate + 100× protease and phosphatase inhibitor cocktail (Cat. No. PPC1010, Sigma-Aldrich) + 100× PMSF (Cat. No. P7626, Sigma-Aldrich)) was added to each sample and transferred into a 2 mL tube prefilled with Triple-Pure High Impact Zirconium Beads (Cat. No. D1032-30, Benchmark Scientific). Then, this was homogenized again using an electrical homogenizer. After the second homogenization, samples were centrifuged for 20 min at 13,000 rpm at 4 °C. The supernatants were then collected and with a dilution of 1000×, the insulin and proinsulin content of the grafts were then measured using Mercodia insulin and Mercodia proinsulin kits. For the complete extraction buffer, we followed a recipe from Abcam: https://www.abcam.com/protocols/elisa-sample-preparation-guide-1 (accessed on 1 January 1998).

### 4.14. Electron Microscopy

The isolated graft was dissected into approximately 1 mm^3^ biopsies and placed in fixative (2.5% glutaraldehyde and 2% paraformaldehyde in 100 mM sodium cacodylate buffer (pH 7.4) (Cat. No. 15960-01, EMSDIASUM) overnight at 4 °C. Samples were then sent to the Electron Microscopy Lab at Nathan S. Kline Institute (http://cdr.rfmh.org/about_facilities.html (accessed on 1 January 2007) for further processing and imaging. The mouse strain was generated by Regeneron for the pancreas sectioning and imaging in Figure 2C.

### 4.15. Statistical Analysis

Data were analyzed using an unpaired *t*-test (for 2-sample comparison) and one-way ANOVA followed by Tukey’s multiple comparison test (for ≥3 samples) (GraphPad Prism 6, GraphPad Software, Inc., La Jolla, CA, USA). All data were plotted as mean ± standard deviation (SD). The differences observed were evaluated as statistically significant when *p*-value was less than 0.05 and were displayed on figures as follows: * *p* < 0.05, ** *p* < 0.01, *** *p* < 0.001, **** *p* < 0.0001.

## Figures and Tables

**Figure 1 cells-12-00903-f001:**
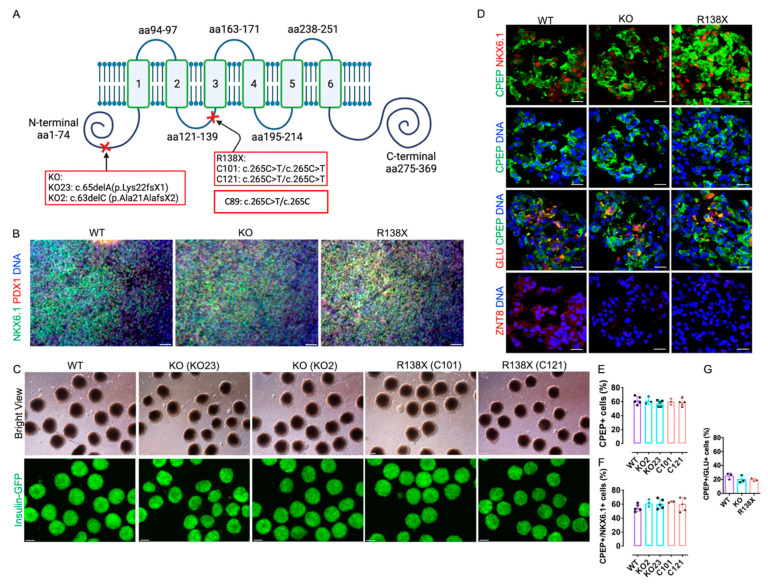
Human pluripotent stem cells with R138X and KO mutation differentiate efficiently to insulin-producing cells. (**A**) Schematic of introduced *SLC30A8* KO mutation (clone KO23: c.65delA/p.Lys22fsX1 and KO2: c.63delC/p.Ala21AlafsX2) and R138X mutation (clone C121: c.265C > T/c.265C > T and C101: c.265C > T/c.265C > T). Adapted from [21]. (**B**) Representative fluorescence picture of stem cell-derived pancreatic progenitor cells on day 12 of differentiation stained for PDX1, NKX6.1, and Hoechst. Scale bar: 50 μm. (**C**) Representative bright field view and fluorescence picture of islet-like clusters derived from WT, two KO clones (KO2 and KO23), and two R138X clones (C101 and C121) on day 27 of differentiation. Scale bar: 100 μm. (**D**) Immunofluorescence of WT, KO, and R138X-derived islet-like clusters stained for C-peptide, NKX6.1, Glucagon, ZnT8, and Hoechst. Scale bar: 20 μm. (**E**–**G**) IF image quantification of C-peptide-positive cells, C-peptide, and NKX6.1 double-positive cells and C-peptide and Glucagon double-positive cells at day 27 of differentiation of WT, KO (KO23), and R138X (C121) stem cells (*n* = 3 for each genotype).

**Figure 2 cells-12-00903-f002:**
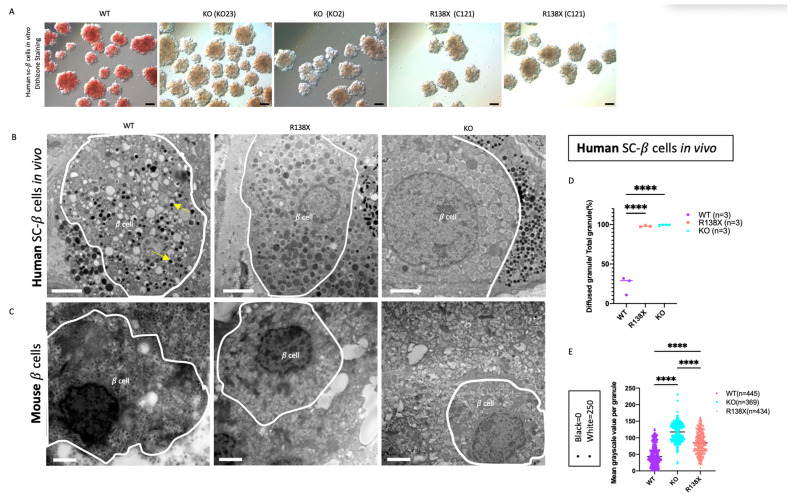
Stem cell-derived beta cells with R138X and KO mutations are zinc depleted and diffused cores are formed in insulin granules. (**A**) Representative bright view pictures of stem cell-derived beta cell clusters with dithizone staining. Scale bar: 100 μm. (**B**) Representative electron microscopy pictures of insulin granules in grafted cells isolated from mice transplanted with WT, R138X, and KO stem cell-derived beta cell clusters for 8 months in vivo. Scale bar: 2 μm. (**C**) Representative electron microscopy pictures of insulin granules in a beta cell of WT mice and R138X mice under chow diet and KO mice under high fat diet. Scale bar: 2 μm. (**D**) Quantification of diffused granules out of total granules in WT, R138X, and KO in percentage: WT (*n* = 3), R138X (*n* = 3) and KO (*n* = 3). One-way ANOVA with **** *p* < 0.0001 for WT vs. R138X and WT vs. KO. (**E**) Quantification of mean grayscale value per granule among WT (*n* = 445), KO (*n* = 369) and R138X (*n* = 434) in (**B**) and not shown. (Black = 0, white = 250; higher grayscale value means lighter in brightness). One-way ANOVA, *p* **** < 0.0001 for WT vs. R138X and WT vs. KO.

**Figure 3 cells-12-00903-f003:**
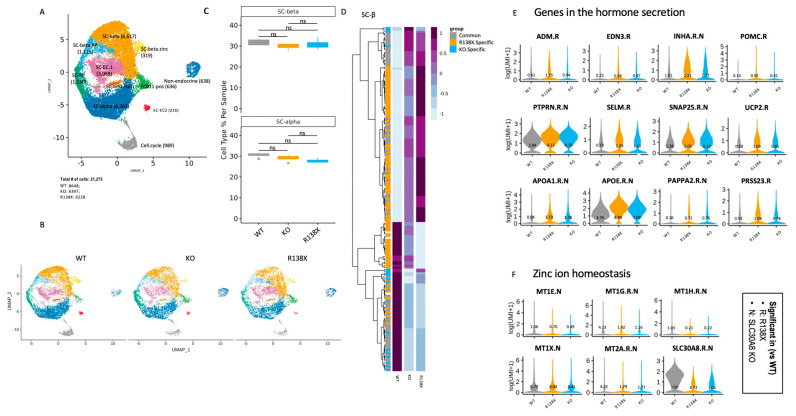
Single cell transcriptome analysis shows disruption of Zn^2+^ homeostasis, and genes enriched in insulin secretion are upregulated in mutants. (**A**) Identified cell populations in stem cell-derived islet like clusters (samples were collected from 4 independent wells of 1 experiment in each condition). (**B**) Identified cell populations in stem cell-derived islet cells separated by genotypes (WT, KO, and R138X). (**C**) Quantification of SC-beta and SC-alpha cell populations of WT, KO, and R138X. Wilcoxon test with ns: not significant. (**D**) The upregulated and downregulated genes in WT, KO, and R138X stem cell-derived beta cells. (**E**) Genes in hormone secretion pathway upregulated in *SLC30A8* mutants (KO and R138X). (**F**) Genes in zinc ion homeostasis pathway downregulated in *SLC30A8* mutants (KO and R138X).

**Figure 4 cells-12-00903-f004:**
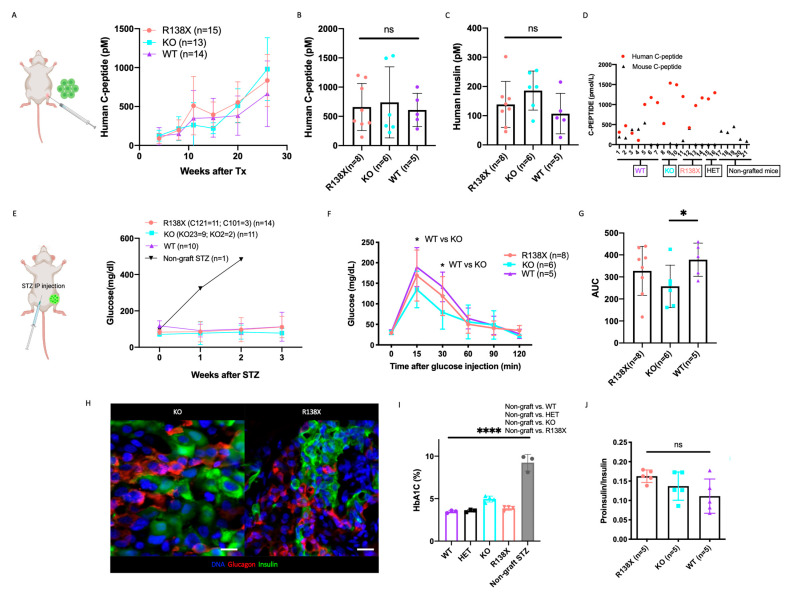
*SLC30A8* KO and R138X mutant stem cell-derived beta cells have comparable glucose regulation compared to WT. (**A**) Human C-peptide secretion was monitored in mice at indicated time points after transplantation with WT (*n* = 14), KO (*n* = 13, including KO2 (*n* = 4) and KO23 (*n* = 9) and R138X (*n* = 15, including C101 (*n* = 4) and C121 (*n* = 11)) stem cell-derived beta cells at fed state. (**B**) The secretion of human C-peptide and (**C**) human insulin were measured in serum of mice transplanted with WT (*n* = 5), KO (*n* = 6, including KO2 (*n* = 2), KO23 (*n* = 4)), R138X (*n* = 8, including C101 (*n* = 3), C121 (*n* = 5)) at 26 weeks after transplantation and before STZ. One-way ANOVA with ns: not significant. (**D**) Serum human and mouse C-peptide were measured in sc-beta cells grafted mice before STZ treatment at fed state. Mouse C-peptide were also measured in non-grafted mice at fed state. (**E**) Blood glucose level was monitored in mice at indicated time points after STZ treatment with WT (*n* = 10), KO (*n* = 11, including KO2 (*n* = 2) and KO23 (*n* = 9), R138X (*n* = 14, including C101 (*n* = 3) and C121 (*n* = 11)) and non-grafted (*n* = 1). (**F**) Glucose tolerance test of STZ-treated mice transplanted with each indicated genotype (WT (*n* = 5), KO (*n* = 6, including KO2 (*n* = 2), KO23 (*n* = 4)), R138X (*n* = 8, including C101 (*n* = 3), C121 (*n* = 5)) in fasting state and at 15, 30, 60, 90 and 120 min after glucose injection. One-way ANOVA with * *p* < 0.05 at 15 min and 30 min (KO vs. WT). (**G**) Quantification of area under curve (AUC) in ipGTT for each indicated genotype. One-way ANOVA with * *p* < 0.05 for KO vs. WT. (**H**) Immunostaining of representative grafts taken from *SLC30A8* KO and R138X with insulin, glucagon, and Hoechst. Scale bar: 20 µm. (**I**) Ratio of proinsulin secretion to insulin secretion of mice transplanted with WT (*n* = 5), KO (*n* = 5) and R138X (*n* = 6) stem cell-derived beta-like cells. One-way ANOVA with ns: not significant. (**I**) HbA1C measured in STZ treated WT (*n* = 3), KO (*n* = 4), and R138X (*n* = 4) sc-beta cell transplanted mice and STZ treated non-grafted mice (data collected from control mice acquired from [27]). One-way ANOVA with **** *p* < 0.0001 (STZ-non grafted vs. WT, KO and R138X). (Only mice who survived STZ treatment were shown in (**B**,**C**,**F**–**J**)).

**Figure 5 cells-12-00903-f005:**
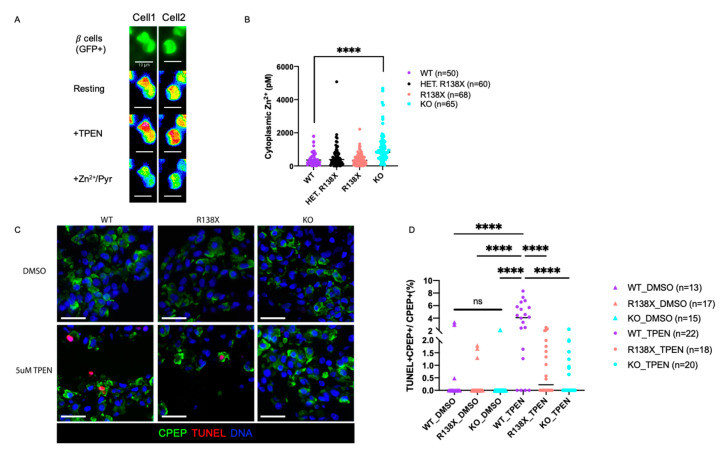
*SLC30A8* mutant sc-beta cells are more resistant to low zinc condition and upregulated cytosolic free Zn^2+^. (**A**) Two representative images (Cell 1 and Cell 2) for cytoplasmic zinc measurement in dispersed insulin GFP-positive cells (maximum ratio upon 5 μM TPEN (zinc chelator) addition and minimum ratio upon Zn^2+^/Pyrithione (zinc ionophore) (Scale bar: 10 μm). (**B**) Zinc concentrations measured in insulin GFP-positive cells from 4 independent replicates: WT (*n* = 50), HET (R138X heterozygous) (*n* = 60), R138X (R138X homozygous) (*n* = 68), and KO (*n* = 65). (**C**) Representative immunostaining of sc-derived cells from WT, KO, R138X with 48 h of DMSO and 5 μM TPEN treatment for C-peptide, TUNEL, and Hoechest. (Scale bar: 20 μm) (**D**) Quantification of TUNEL, C-peptide double positive cells out of total C-peptide positive cells (%) among WT, R138X, and KO lines. One-way ANOVA, *p* **** < 0.0001 for the indicated groups.

## Data Availability

Single-cell transcriptome data of stem cell derived cell clusters (WT, R138X and KO) are deposited in Gene Expression Omnibus GEO: GSE166641.

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
