# Peer review of "ZnT8 Loss of Function Mutation Increases Resistance of Human Embryonic Stem Cell-Derived Beta Cells to Apoptosis in Low Zinc Condition"

_cells, 2023, doi:10.3390/cells12060903_

Round 1

Reviewer 1 Report

This manuscript by Sui et al. investigates the effects of mutations in the human SLC30A8 gene using human pluripotent stem cells as a model system. The authors generated several lines harboring either null or truncating mutations. They assessed the impact of these mutations on pancreatic endocrine cell development and beta cell function. While these mutations didn’t impair the efficiency of differentiation into insulin-producing cells, they observed some changes in insulin granules after transplanting the cells into mice. They also demonstrated that the mutant cells have increased resistance to apoptosis when extracellular zinc is limiting. This manuscript complements previously published studies reporting on the role of SLC30A8 in human beta cells.

Major concern:

Ma et al. (Nature Communications, 2022) and Dwivedi et al. (Nature Genetics, 2019) both reported that ZnT8 loss of function in human cells improves glucose-stimulated insulin secretion. In contrast, the authors of this manuscript claim that insulin secretion in SLC30A8 mutant sc-beta cells is comparable to WT. This claim is based on in vivo measurements of human C-peptide secretion in the fed state, which is quite noisy. In vitro analysis of insulin secretion by WT and mutant sc-beta cells in the presence of high concentrations of glucose would help settle this question.

Minor concerns:

Figure 1B – The expression of each transcription factor is hard to see. The figure would be improved by showing the individual channels for PDX1 and NKX6.1.

Figure 2C –Please provide more details on the mice used to generate this data.

Figure 3 and S2 – Analysis of the data would be easier if the color code for the different populations was the same for all figures.

Figure S2 – Please include expression of NKX6.1.

Figure 3A - What are the differences between SC-beta, SC-beta.PP, SC-beta.1, and SC-beta.zinc? Is SC-beta.zinc of subset of SC-beta that has been previously described?

Figure 3C and 3D – which clusters are included in ‘sc-beta’?

Figure 3D – Please include the list of DE genes either in the figure or in supplementary.

Figure 3E-F – The differences between groups are hard to see. Is there a better way to show this data? Maybe showing the values for individual cells?

Can the authors also make the R/N legend bigger/clearer?

What are the p values? Please include data in supplementary.

Figure S3A – Use different colors for each group of transplanted mice.

Figure S3C – Were non-transplanted controls included in the STZ experiment?

Figure S3F – The numbers are not readable. And why is there data for only three mice? The authors should also indicate the levels of c-peptide measured in the mice that are being imaged (or indicate which mice from Figure S3A).

Figure S3G and H – when were the grafts harvested? Why are there only six mice represented in the graph?

Figure S3K – indicate number of replicates. Why is there no data on secretion in high glucose?

Figure 4F – How was the ratio proinsulin/insulin measured?

Please provide a table with the sequences of the gRNAs and ssDNA template used.

Reviewer 2 Report

In this study, the authors generated hESC lines carrying R138X mutation to understand the role of ZnT8 gene in the development and function of pancreatic beta cells, because this mutation is known to be associated with diabetes. The gene-edited hPSC lines were differentiated into pancreatic beta cells and evaluated the effect of the mutation on the development and function. Their main findings showed that loss of ZnT8 has a protective effect against apoptosis under low zinc conditions. Although the authors performed several experiments, the manuscript lack the novelty. My major comments are as follows:

1)    The major concern of this manuscript is the novelty of the obtained results. There are two important articles published in 2020 and 2022 (PMID: 31676859PMID: 31676859covered the same topic extensively and obtained the same results. Therefore, this manuscript does not add to the field of stem cells/diabetes, because it does not bring any new findings.

2)    All the findings reported under the “Results” section have been previously reported (Ma et al. 2022 and/or Dwivedi et al. 2020), including: I) Human embryonic stem cells with SLC30A8 mutations exhibit normal differentiation potential towards insulin-producing cells; II) Sc-beta cells carrying SLC30A8 mutations have zinc-depleted secretory granules; III) Insulin secretion in SLC30A8 mutant sc-beta cells is comparable to WT; IV) SLC30A8 mutant sc-beta cells are resistant to apoptosis in low-zinc conditions. This indicates that the authors did NOT bring any new findings. 

-       Ma et al. 2022 (PMID: 35842441) established three independent SLC30A8 knockout (KO) hESC (HUES8 and MEL1) lines were established by CRISPR/Cas9-based genome editing. Those KO hESC lines have been used to examine the effect of ZnT8 loss on pancreatic beta cell development and maturation has been investigated.

-       Dwivedi et al. 2020 generated heterozygous iPSC lines carrying the same mutation, the SLC30A8-p.Arg138* andthey knocked down SLC30A8 using siRNA.

3)    The lack of dense-core zinc positive insulin granules in SLC30A8 deficient beta cells have been reported by Ma et al., 2022.

4)    The justifications provided in the “Discussion” about the difference between their study and Ma et al. 2022 study do not bring new findings. 

5)    In the “Discussion”, the author mentioned that “Here we show that the effect of loss of function mutations in SLC30A8 locus on beta cell development and on the function of differentiated beta cells can be investigated using a stem cell model. This has been shown in the previous studies.

6) The Summary in the "Discussion" does not add any new results compared with the previous two studies.

Reviewer 3 Report

The paper by Sui et. al describe the role of the rare SLC30A8 mutation encoding a truncating p.Arg138* variant (R138X) in zinc transporter 8 (ZnT8) in the generation and function of human stem cell derived beta cells and suggest a protective role of mutated beta cells to apoptosis under low zinc conditions.

The paper is well written and all the data are supported by appropriate, detailed and convincing experiments.

Despite this, there are still some points that should be clarified and which prevent me from accepting the manuscript in its current version:

- It is not clear why all the experiments were not performed also with the heterozygous line C89. Only some results are present in some final sections of the results or in the supplementary files. I believe that the results obtained with this cell line should also be reported together with those of the KO and homozygous lines in each figure of the paper.

- lines 194-196 which refer to figure 3E report a lower expression of the MT genes which however is not visible in the figure in the current version. The data should be represented to make this difference more evident.

- the percentages of c-Pep+/NKX6.1+ cells obtained by FACS (Fig S1D) and by IF quantifications are very different (from an average of 40% to 60%), the authors should explain and go into detail of this difference. Furthermore, Figure S1D does not reflect the quantification in Figure S1E.

- figure S3C shows the glycaemia of the transplanted mice after treatment with STZ. The glycemic profile of non-transplanted mice treated with STZ should also be included as an internal control in this figure to further validate the efficacy of the transplanted cells in maintaining normoglycemia. This figure could then be integrated into Figure 4.

- the percentage of TUNEL+ cells in WT cells after treatment with TPEN for 48h is, even if significant, rather low. The authors should insert a positive apoptosis control to validate the specificity of the resistance to apoptosis and to understand if that small percentage of death in WT cells could have a biological significance.

Some minor points:

- the figure legends of figure 3E and F are reversed

- in line 203 is cited the figure S4A, which is mentioned before the figure S3, therefore they should be interchanged.

Reviewer 4 Report

In this article, Sui and colleagues used human pluripotent stem cells to model beta cell development and study ZnT8 hypomorphic variants and knockouts. They found no effects on beta cell differentiation or in mature function after transplantation. The authors used several edited PSC lines, and a wide array of methods to reach these conclusions. The authors also report  transcriptional differences in genes associated with Zn2+ homeostasis and describe increased resistance to Zn depletion-induced apoptosis in ZnT8 KO and hypomorphic lines.

While studies on these variants are not novel, and others have also previously used similar models to try to unravel the molecular mechanism underlying the "decreased T2D risk" observed in humans with the R138X variant, Sui and colleagues report differences with previous published articles, particularly Ma et al., and  Dwivedi et al. The article is well written and the discussion enjoyable. Besides, the dataset generated by the authors will be an important asset for future studies that will follow, since in my view, the link between R138X (or other hylomorphic variants as found in GWAS nearly 15 years ago), is far for been mechanistically determined. 

Minor comments:

- The effects on cytosolic levels of Zn observed by authors are not but not observed in other models (including Mitchell et al, which describe a diametrically opposed result). Could this be a consequence of differences of Zn levels on experimental models? Is the 10 µM ZnSO4 used by authors comparable to the levels used in other models, or is it a biologically relevant level? Could the authors comment on this point?

- Fig5. legend title: I would suggest to change "due to" by "and", since the authors have not sufficiently proven this.

Some proofreading is required, examples:

- Fig1. c.63celC, c.65delC (inconsistencies between figure and text)

- Fig legend 2: Title is wrong

- Fig3. interchanged E/F legends

- Line 162 "highest expression IDENTIFIED IN ..."

- Line 420-421 please rewrite

Round 2

Reviewer 1 Report

The authors have addressed most of my concerns.

Reviewer 2 Report

Although the authors tried to respond to my questions, but their responses are not convincing. The novelty of this manuscript is the major concern. As expert in the field, I do not see how the data presented will add to the field in the presence of two stronger published articles covered the same topic with clearer results (PMID: 31676859; PMID: 31676859).

Reviewer 3 Report

The authors answered my questions convincingly, the manuscript can be considered by the editors for publication